# A Grading Method of Ginseng (*Panax ginseng C. A. Meyer*) Appearance Quality Based on an Improved ResNet50 Model

**Dongming Li** [1], **Xinru Piao** [1] , **Yu Lei** [1], **Wei Li** [2], **Lijuan Zhang** [3] and **Li Ma** [1,*]

1 College of Information Technology, Jilin Agricultural University, Changchun 130118, China
2 College of Chinese Medicinal Materials, Jilin Agricultural University, Changchun 130118, China
3 College of Computer Science and Engineering, Changchun University of Technology, Changchun 130012, China
* Correspondence: mali@jlau.edu.cn; Tel.: +86-13-5008-83257

**Abstract:** In the academic world, ginseng (*Panax ginseng C. A. Meyer*) has received much attention as the most representative element of Chinese medicine. To address the lack of traditional algorithms in the identification of ginseng appearance quality and further improve the manual identification on ginseng, we propose a grading method of ginseng appearance quality based on deep learning, taking advantage of the benefits of deep learning in the image identification. Firstly, we substituted *Leaky ReLU* for the conventional activation function *ReLU* to enhance the predictive power of the model. Secondly, we added an ECA module to the residual block, which allowed attention to be focused on the input object to capture more precise and detailed features. Thirdly, we used the focal loss function to solve the problem of an imbalanced dataset. Then, the self-constructed dataset was processed with data enhancement and divided into four different classes of ginseng. The dataset was trained on a model with transfer learning to finally obtain the best model applicable to the identification of ginseng appearance quality. The experiments showed that, compared with the classical convolutional neural network models VGG16, GoogLeNet, ResNet50 and Densenet121, the proposed model reported the best performance, its accuracy in the test set was as high as 97.39%, and the loss value was 0.035. This method can efficiently classify the appearance quality of ginseng, and has a significant value in the field of ginseng appearance quality identification.

**Keywords:** appearance quality identification of ginseng; deep learning; attention mechanism; activation function; loss function

## 1. Introduction

Ginseng (*Panax ginseng C. A. Meyer*) is the dried root and rhizome of ginseng of the family Wujia, which has the effects of tonifying the kidneys, calming the mind, nourishing the brain, brightening the eyes, and beneficial to intellectual development [1]. It is referred to as the "king of all herbs" since it is a priceless Chinese herb and tonic [2]. The appearance of particular ginseng features is the key factor used in the traditional method of ginseng identification to assess its quality. The literature [3] also demonstrates that there is a correlation between the appearance of ginseng's characteristic traits and its main chemical components. Empirical identification of ginseng quality based on "identifying the appearance and shape to discuss the quality" is a scientific method [4–6] that is particularly effective. However, it has shortcomings. To identify a ginseng specimen, a connoisseur must first carefully observe all aspects of the ginseng plant under exam, which requires a lot of time and effort. Secondly, ginseng with high economic and therapeutic value may be damaged, since the manual identification process has a tendency to harm the plant. In the end, manual recognition is subject to some degree of subjective influence. This frequently leads to inconsistent identification criteria and incorrect identification outcomes [7]. At present, there is a lack of sufficient attention to the study of ginseng characteristics. The 2020 edition of the Chinese Pharmacopoeia also includes descriptions and specificities of ginseng phenotypic features;

however, in practice, the general public will still not be able to tell ginseng quality by its appearance. With the rapid advancement of computer technology, artificial intelligence technology has slowly begun to be integrated in traditional Chinese medicine identification. The "Fourteenth Five-Year Plan for the Development of Chinese Medicine Informatization" announced in 2022 makes it very obvious that informational means must be used to advance the modernization of Chinese medicine.

Deep learning is close to the way humans learn by mimicking the neural networks of the human brain, using a hierarchical network model structure to gather information about the appearance and sound of things, to perceive and understand them, and to generate appropriate behaviors. Its development in the field of computer vision is more complete than in other areas and has produced some outcomes [8]. LECUN et al. [9] proposed LeNet, which is the most representative convolutional neural network and consists of a Conv layer, a pooling layer, and a fully connected layer. When the network was initially used to accurately classify handwritten datasets, it established the stage for the later development of convolutional neural networks. With the AlexNet [10] network winning the ImageNet (large visualization database) competition in 2012, a number of classical deep convolutional neural networks have emerged, such as VGG [11], GoogLeNet [12], ResNet [13], DenseNet [14], etc. They outperform conventional approaches in picture identification and offer greater advantages. These networks are already extensively utilized in the fields of Chinese medicine and plant identification, despite the fact that they have not yet been employed to identify the appearance and quality of ginseng. Dyrmann et al. [15] adopted a method for identifying plant species in color images using convolutional neural networks and were able to identify 22 weed and crop species with an accuracy of 86.2%. Lee et al. [16] designed a deep learning method for the quantitative discrimination of leaves by gathering data on leaf features and analyzing them through convolutional neural networks and deconvolutional networks. The results of the study demonstrated that using deep learning methods can further increase the identification efficiency of plant classification based on the leaf. Liu et al. [17] applied the Inception structure and introduced dense connectivity to successfully achieve the classification and identification of six grape diseases, and the final model accuracy was 97.22%. Liu Wei et al. [18] combined Xception and DenseNet to propose the new image identification model DxFusion, which was able to accurately identify 60 Chinese herbs. Li Dongming et al. [19] used a deep learning strategy with the residual network and densely linked network to eventually identify and categorize five distinct origins of the Saposhnikovia divaricate.

Ginseng appearance quality identification is a fine-grained classification task. Therefore, considering the above research analysis, we attempted to apply deep learning techniques to the field of ginseng appearance quality identification and enhance ResNet50 according to the characteristics of ginseng. We finally propose an algorithm that can be effective, quick, and precise for grading the appearance and quality of ginseng.

## 2. Data Pre-Processing

### 2.1. Dataset Construction

After analyzing various ginseng varieties, we chose to employ white ginseng (a product made from fresh garden ginseng that was cleaned and dried or dried in the sun) as the experimental subject due to the complex morphological structure of the herb and market demand. The experimental data were gathered in September 2021 at the School of Chinese Herbal Medicine, Jilin Agricultural University, from the same batch of white ginseng. To facilitate the image collection, the white ginseng batch was assigned a serial number and scored by experts in accordance with the guidelines for white ginseng in the document "Group Standard for Ginseng of Jilin Daoji Herbs" which was released by the Tonghua Ginseng Industry Association on 10 September 2021 (as in Table 1). Ginseng of a standard lower than the principal, first-class, and second-class was rated as inferior.

The specimens were then photographed according to their ratings by using a small HD folding studio box (Sutefoto., Guangdong, China). A mobile phone camera (Apple.,

Cupertino, CA, USA) was positioned at the top of the studio box, perpendicular to the ginseng, at a height of 40 cm, guaranteeing that all samples were essentially photographed in the same position. The photographs of the white ginseng were gathered from various angles and backgrounds, and each image had a resolution of 1024 × 1024 pixels, with distinct image details. A total of 549 images of the principal white ginsengs, 790 images of the first-class white ginsengs, 600 images of the second-class white ginsengs, and 290 images of the inferior white ginsengs were saved after numerous photos were shot and gathered.

**Table 1.** Scoring criteria.

| Project | Principal | First-Class | Second-Class |
|---|---|---|---|
| Rutabaga | Complete with rutabaga and ginseng *(Panax ginseng C. A. Meye)* fibrous roots | The rutabaga and ginseng fibrous roots are more complete | Rutabaga and ginseng with incomplete fibrous roots |
| Surface | Yellowish white or greyish yellow, without water embroidery or guttering | Yellowish white or greyish yellow, or with water rust or guttering | Yellowish white or greyish yellow with rust and guttering |
| Breakage Scar | None | Mild | Have |
| Branching out | No small ginseng or ginseng whiskers are allowed to be caught by those with branching roots, no tied tails or lightly tied tails or tied tails | | |
| Section | Section pale yellowish white, pink | | |
| Texture | Hard, powdery, no hollow | | |
| Odor | Unique aroma, taste slightly bitter, sweet | | |
| Main Root | Cylindrical | | |
| Insects Mildew Impurities | None | | |

## 2.2. Data Enhancement

When the number and quality of the data were higher during the deep learning training phase, the model was more generalizable. As a result, the upper limit of the model learning was directly determined by the data. During the collection of the data, most of the photographs had difficulty in reproducing the full scene of the environment in which the samples were positioned, and the number of photographs taken was small. Because of this, we needed to perform data enhancement on the collected photos [20]. In this paper, two types of enhancement were adopted: offline enhancement and online enhancement.

The offline enhancement method applies random rotation, random adjustment of contrast, and random adjustment of brightness to expand the dataset. Following offline augmentation, the data were increased to 6131 frames. Information on the dataset is shown in Figure 1.

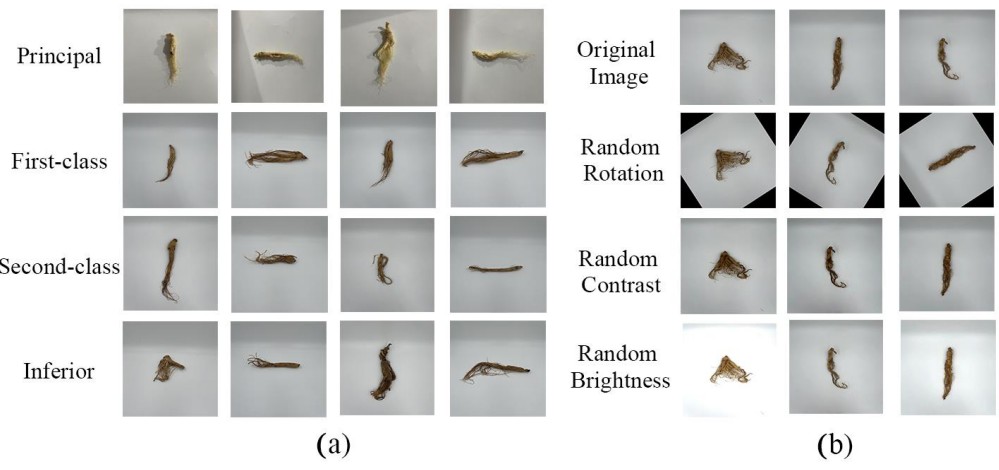

(a)             (b)

**Figure 1.** Ginseng *(Panax ginseng C. A. Meye)* dataset. (**a**) Different levels of ginseng images; (**b**) Image enhancement example diagram.

The online enhancement method uses the PIL (Python Image Library) module in the Python image library to uniformly crop a given image to 256 pixels by 256 pixels before each training round; then the images are processed using the center crop method, and finally the normalization is completed. This could indirectly increase the amount of training data by increasing the diversity of the image samples used in the training process.

### 2.3. Dataset Partitioning

To eliminate the serendipity of the experimental results, all training in this paper was conducted by using the five-fold cross-validation method. A randomly selected 80% of the dataset was used as the training set, and 20% of it as the test set. That is, 1225 images were used as validation data each time, while 4906 images were extracted from the training set as training data, and the validation data were not repeated. The final results were the average of five experiments, and the breakdown of the data for each class is shown in Table 2.

**Table 2.** Dataset partition.

| Level | Training Set Image | Verification Set Image |
|---|---|---|
| Principal | 1225 | 306 |
| First-class | 1396 | 348 |
| Second-class | 1589 | 397 |
| Inferior | 696 | 174 |

## 3. Building the Network Model

### 3.1. Resnet50 Model

He et al. proposed ResNet in 2015 [13], in which the residual block is an important structure. It enhances the transfer of features by introducing shortcut connections in the convolutional neural network, so that the next layer contains more information about the image. The + residual block consists of a Convolution Layer (Conv), a Batch Normalization Layer (BN), and a Rectified Linear Units Layer (*ReLU*). The structure of the residual block used by ResNet50 is shown in Figure 2. Here, $x$ is the input to the network, and $F(x)$ represents the output after three Conv layers of processing. As the depth of the neural network increases, the image information in the feature map decreases layer by layer. Therefore, it is merged with the original output through the mapping of shortcut connections before being delivered to the subsequent layer $F(x) + x$ The issue that typical deep learning algorithms cause the network to burst and disappear as the number of layers grows, making the model difficult to converge, is resolved by this operation. The main branch contains three Conv layers, the first being a $1 \times 1$ Conv layer to compress dimensionality, the second a $3 \times 3$ Conv layer, and the third a $1 \times 1$ Conv layer to reduce dimensionality. This greatly maintains accuracy and reduces the computational effort [21].

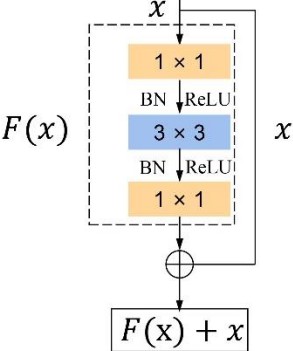

**Figure 2.** Residual block.

### 3.2. Improving the Resnet50 Model

#### 3.2.1. Using the Leaky ReLU Activation Function

Using activation functions in neural networks can increase the non-linear variability of the neural network model. The traditional ResNet uses a *ReLU* activation function. It is characterized by fast computing speed and good performance. Its function expression is shown in Equation (1).

$$ReLU(x) = \begin{cases} 0, x \leq 0 \\ x, x > 0 \end{cases} \tag{1}$$

As can be seen in Figure 3, the gradient of this interval is constant when $x$ of the *ReLU* function is greater than 0, thus alleviating the gradient disappearance problem. However, when the input value is negative during training, the function will reach hard saturation, resulting in the weights not being updated, i.e., the neuron death phenomenon.

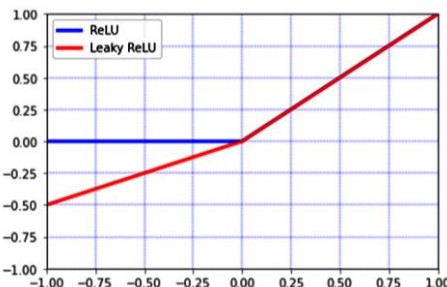

**Figure 3.** Comparison of the *ReLU* and *Leaky ReLU* functions.

To address this problem, we replaced the *ReLU* activation function with a *Leaky ReLU* [22]. The *Leaky ReLU* function has the characteristics of being able to achieve hard saturation, fast computation, and quick convergence compared to the conventional activation functions Sigmoid and Tanh. Its expression is shown in Equation (2).

$$Leaky\ ReLU(\mathrm{x}) = \begin{cases} x, x \geq 0 \\ \alpha x, x < 0 \end{cases} \tag{2}$$

where $\alpha$ is the value of the gradient, the default setting being $\alpha = 0.5$. The capacity of the *Leaky ReLU* function to preserve the gradient when the input information is less than 0 gives it an advantage over the *ReLU* function, allowing the parameters to continue to be updated, as can also be seen from the comparison chart in Figure 3. By doing this, the network's interpretability will be enhanced, and any data loss will be prevented.

#### 3.2.2. Adding an Attention Mechanism

The use of an attention mechanism in convolutional neural networks has been much favored in recent years, as it can substantially improve a network performance by refining the feature mapping [23–25]. The channel attention mechanism has demonstrated the most potential for enhancing network performance among the available attention mechanisms. One of its representative networks is the Squeeze and Excitation Network (SENet) that incorporates the SE module (Squeeze and Excitation) [26]. However, the feature extraction process's dimensionality reduction has negative effects on channel attention prediction and ineffectively captures the dependencies between all channels. Therefore, Wang et al. then proposed a new efficient channel attention network, i.e., the ECA-Net (Efficient Channel Attention for Deep Convolutional Neural Networks), in 2019 [27]. The structure of the ECA module is shown in Figure 4. Without dimensionality reduction, the ECA module has the ability to adaptively choose one-dimensional convolutional kernels. Here is how it works.

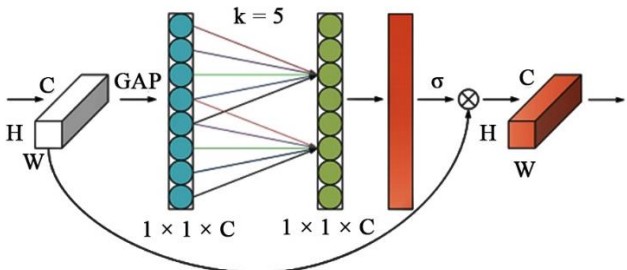

**Figure 4.** Structure of the Efficient Channel Attention Module.

(1) Input features: Given an input feature image $X$ of size $H \times W \times C$, the input image is subjected to Global Average Pooling (GAP) to obtain all the feature information without dimensionality reduction.

(2) Calculation: The one-dimensional convolution operation with a convolution kernel of size $k$ is used to efficiently complete the cross-channel information interaction and obtain the weights of each channel $\omega$, as shown in Equation (3).

$$\omega = \delta(C1D_k(y)) \tag{3}$$

where $\delta$ represents the sigmoid activation function, and $C1D$ represents the one-dimensional convolution. The number of channels $C$ is proportional to the one-dimensional convolution with kernel $k$, as shown in Equation (4).

$$C = 2^{(\gamma * k - b)} \tag{4}$$

The final kernel size $k$ can be determined adaptively using Equation (5).

$$k = \left| \frac{\log_2(C)}{\gamma} + \frac{b}{\gamma} \right|_{odd} \tag{5}$$

where $t$ is the nearest odd number to $|t|_{odd}$, $b$ is 1, and $\gamma$ is 2.

(3) Output features: The weights of each channel are obtained using the sigmoid function, and the input feature maps are channel-weighted to finally obtain the feature maps under different weights.

Since backpropagating the feature information is vulnerable to gradient dissipation close to the input layer, it is challenging to boost the network model's efficiency. The majority of ginseng plants share a similar shape and have fine textures and dense roots, which can affect their identification after downsampling and makes it difficult to extract detailed features from the network. Therefore, we improved the original residual block by adding the ECA module before the feature overlay, as shown in Figure 5. With the addition of the ECA module, the network enhanced the learning of channel attention features for each Conv block, and focusing the attention on hard-to-identify ginseng images allowed the features to be reused. As a result, the network model performed significantly better.

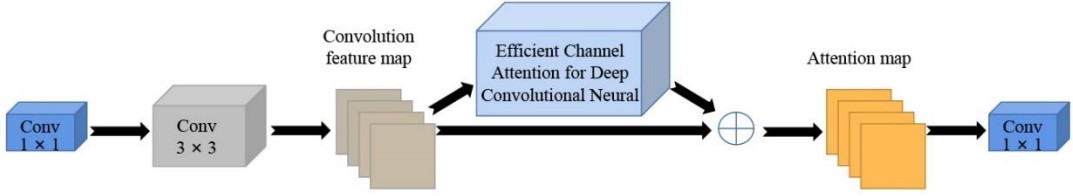

**Figure 5.** Adding the residual block structure of the ECA attention mechanism.

### 3.2.3. Introduction of a Focal Loss Function

The focal loss function (Focal Loss) [28] is the main solution to the problem of the existence of imbalances in the classification dataset and of mining difficult samples. It is based on an enhancement of the cross-entropy loss function [29]. Ordinary cross entropy states that for positive samples, the output probability increases, and the loss decreases; for negative samples, the output probability decreases, and the loss increases. In this case, the loss function improved slowly over the iterative process of a large number of simple samples and might not be able to optimize to the optimum. The original base was increased by the adjustment factor $\gamma(\gamma > 0)$. Its formula is shown in Equation (6); by reducing the loss of easy-to-classify samples, it concentrates more on complex, incorrectly classified samples. When $\gamma$ is 0, we have the cross-entropy loss function.

$$L_{fl} = \begin{cases} -(1-y')^{\gamma} \log y', y = 1 \\ -y'^{\gamma} \log(1-y'), y = 0 \end{cases} \tag{6}$$

In addition to this, a balancing factor $\alpha$ was introduced to compensate the unequal distribution of positive and negative samples; its formula is given in Equation (7).

$$L_{fl} = \begin{cases} -\alpha(1-y')^{\gamma} \log y', y = 1 \\ -(1-\alpha)y'^{\gamma} log(1-y)', y = 0 \end{cases} \tag{7}$$

Due to the minimal number of inferior samples in the constructed ginseng dataset and the lack of distinctive features between different ginseng species, to reduce the weight of the negative samples in the training, a focal loss function was introduced; this allowed the model to obtain more accurate classification results.

### 3.2.4. Structure of the Ginseng Appearance Quality Grading Model

The classical ResNet consists of three parts: the input, the convolution, and the output. First, the data enter the network and go through the input section. Then, they go through the intermediate 1, 2, 3, and 4 Conv layer sections. Finally, the data go to the averaging pooling and fully connected layers to acquire the result. In this study, the original ResNet50 was reconstructed using the three suggested strategies for improvement. The enhanced network structure is shown in Figure 6.

The input feature map of this network first passes through a 7 × 7 Conv layer, preserving the image's original features over a substantial area. Then, after the BN (BatchNormalization) layer and the *Leaky ReLU* activation function, the features are extracted from the feature map using a 3 × 3 maximum pooling layer and compressed into a feature map with a channel count of 64. The feature map enters four layers in turn. Each layer consists of a Conv block and a variable number of ID (Identity) blocks. One Conv block and two ID blocks make up the first layer, one Conv block and three ID blocks make up layer 2, one Conv block and five ID blocks make up layer 3, and one Conv block and two ID blocks make up layer 4. Each Conv block and ID block adds the ECA module as indicated in Figure 6b,c. Eventually, a feature map with a channel count of 2048 is produced by successively passing the shallow spatial information and the underlying semantic information. The constructed model employs a global average pooling layer to optimize the network structure and adds a discard layer before the fully connected layer to prevent overfitting. The feature map is finally sent to the fully connected layer prediction to obtain the classification result.

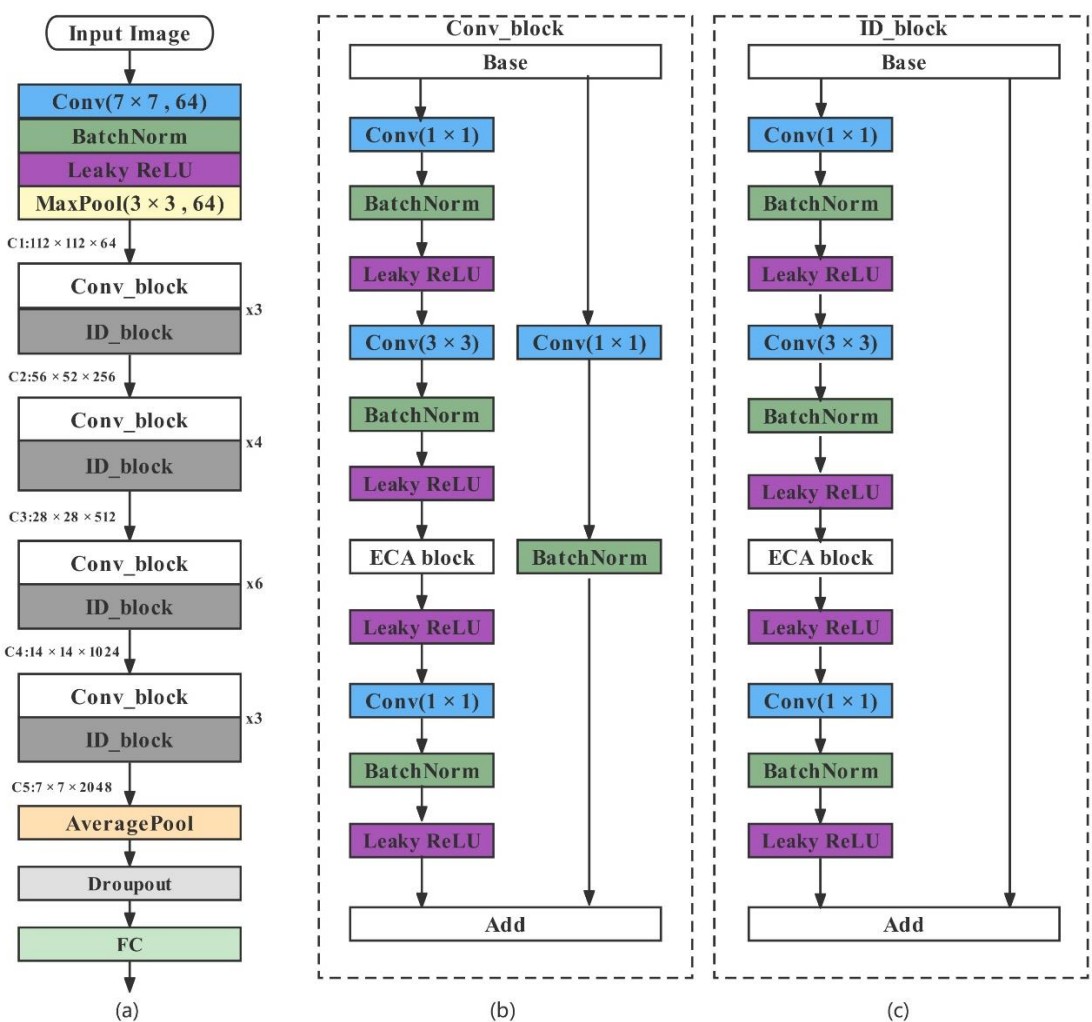

**Figure 6.** Structure of the improved model. (**a**) Backbone structure; (**b**) Conv_block; (**c**) ID_block. Note: Conv is Conv Layer; BatchNorm is Batch Normalization; ID_Block is identity block; Conv_block is convolution block; FC is full connection; ECA block is Efficient Channel Attention block.

### 3.3. Model Training for Transfer Learning

Transfer learning [30] involves moving pre-trained model parameters to a newly constructed model and sharing the newly learned knowledge with the new model in order to accelerate and optimize the learning speed of the model and bring it to convergence in the shortest possible time. Due to the fact that the model training used a supervised learning approach, a substantial amount of data samples were needed once the ginseng appearance quality grading model had been constructed. We had a limited number of samples, which made it challenging to set up the deep network training. However, the use of transfer learning methods could effectively improve the accuracy and generalization of the model. First, the constructed ginseng appearance quality grading model was initialized by loading the dataset from ImageNet. After that, the trained weights were transferred to the enhanced Conv layer. Finally, the ginseng dataset was loaded into the network, and then all layers aside from the fully connected layer were frozen and continuously fine-tuned for training until the optimal model was obtained [31].

## 4. Experimental Validation and Analysis of the Results

### 4.1. Experimental Setup and Analysis of the Results

We implemented our approach based on PyTorch. The processor of the experimental workstation was Xeon 4210 (8-core 2.45 GHz) (Intel., Santa Clara, CA, USA); the memory

was 64 G. the GPU was NVIDIA GeForce GTX 1080 ti (NVIDIA., Santa Clara, CA, USA); the running memory was 11 GBRAM. The software experimental configuration environment was Ubuntu 16.04; Python 3.7.0; Pytorch 1.10.1; CUDA 10.2. The specific parameter settings in the experiment are shown in Table 3.

**Table 3.** Model parameter settings.

| Parameter | Set Up |
|---|---|
| Optimizer | Adam |
| Learning rate | 0.0001 |
| Weight decay | 0.0001 |
| Batch size | 32 |
| Epoch | 50 |
| Loss function | Focal Loss |

To confirm the validity of the model, we compared the improved model with the classical convolutional neural network models VGG16, GoogLeNet, ResNet50, and DenseNet121. The accuracy rates increased by 3.76, 2.61, 2.45, and 1.88 percentage points, and the loss values decreased by 0.091, 0.042, 0.031, and 0.022 percentage points, as shown in Table 4. The comparison graph in Figure 7 shows that the improved model showed the highest recognition accuracy and the lowest loss value when compared to the other models as well as the shortest training time per round, and could quickly converge to find the optimal value; it also reached convergence quickly to find the optimal value. This demonstrated the superior performance of the model, indicating that it would provide a valuable reference for the subsequent application in ginseng appearance quality identification.

**Table 4.** Comparison of the experimental results of various models.

| Model | Accuracy/% | Loss | Convergence/Epoch | Training Time per Round/s |
|---|---|---|---|---|
| VGG16 | 93.63 | 0.126 | 22 | 20 |
| GoogLeNet | 94.78 | 0.077 | 18 | 18 |
| ResNet50 | 94.94 | 0.066 | 15 | 22 |
| DenseNet121 | 95.51 | 0.057 | 12 | 25 |
| Our model | 97.39 | 0.035 | 5 | 19 |

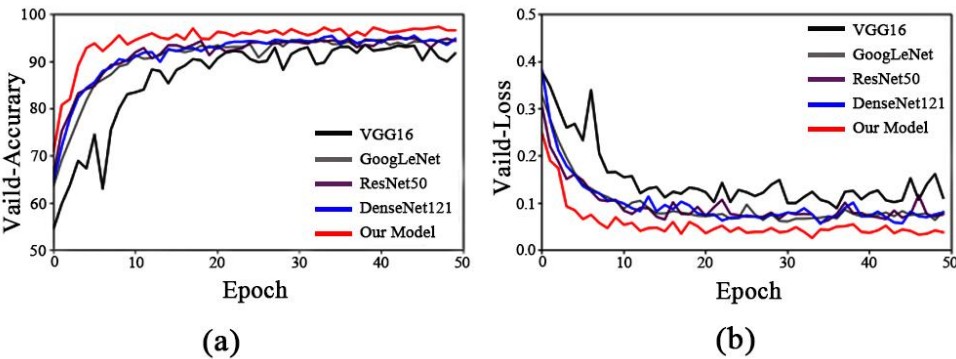

**Figure 7.** Identification results of each mode. (**a**) Model accuracy; (**b**) model loss.

### 4.2. Impact of Attention Mechanisms on the Model Performance

In our experiments, we found that the addition of different attention mechanisms also had some effects on the model's performance when all other factors were equal. As shown in Table 5, the model improved its recognition accuracy by 0.74, 1.31, and 1.72 percentage

points, and the loss values decreased by 0.002, 0.02, and 0.021 percentage points, respectively, with the addition of the *Leaky ReLU* activation function alone and then the addition of the SE channel attention mechanism [26], the CBAM (Convolutional Block Attention Module) convolutional attention mechanism [32], and the ECA efficient channel attention mechanism. While the loss values decreased by 0.002, 0.02 and 0.021 percentage points, the recognition accuracy increased by 0.74, 1.31 and 1.72 percentage points, respectively. In contrast, the addition of the SK (Selective Kernel) attention mechanism [33] resulted in a 2.857 percentage point decrease in identification accuracy and a concomitant 0.019 increase in loss value. The results showed that the ECA mechanism had a more significant role in improving the performance of the network model than the other attention mechanisms. However, the results of incorporating the SK attention mechanism also demonstrated that the introduction of an attention mechanism into the network model does not guarantee an accurate identification; therefore the choice of the attention mechanism needs to be made appropriately for different tasks.

**Table 5.** Comparison of the results obtained with different attention mechanisms.

| Model | Accuracy/% | Loss |
| --- | --- | --- |
| No attention | 95.67 | 0.056 |
| SE | 96.41 | 0.054 |
| CBAM | 96.98 | 0.036 |
| SK | 92.82 | 0.075 |
| ECA | 97.39 | 0.035 |

Note: SE is Squeeze and Excitation; CBAM is Convolutional Block Attention Module; SK is Selective Kernel.

Furthermore, the visualization images obtained by the tool Grad-Cam [34] provided a more intuitive view of the regions of the network's attention to the feature maps before and after the addition of the ECA mechanism. Figure 8 clearly demonstrates how effectively the ECA mechanism module we introduced could capture the ginseng features. It overcame the conflict between performance and complexity by using fewer parameters. To enhance the feature representation of the network model, it efficiently made use of the ability of convolutional neural networks to capture details about neighborhoods. The complex texture of ginseng and slight variations in appearance could also be used to effectively address the negative impact on experimental accuracy.

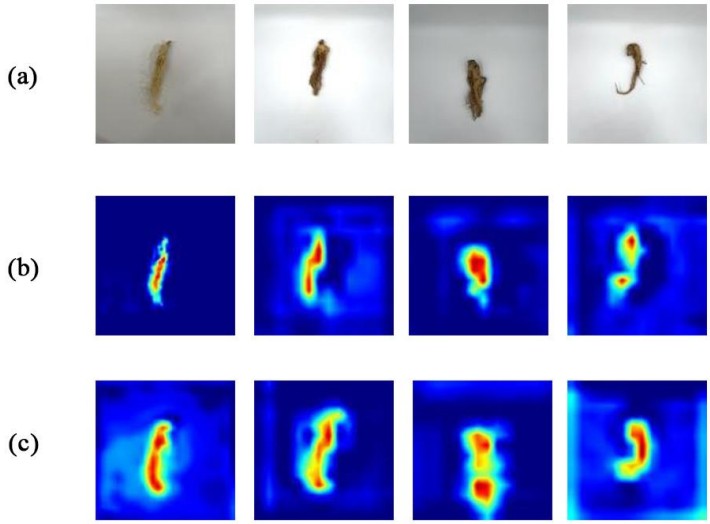

**Figure 8.** Visualization results of the thermal characteristic diagram of the new network before and after adding the ECA module. (**a**) Input image; (**b**) image before joining the ECA module; (**c**) image after adding the ECA module.

### 4.3. Effect of the Activation Function on the Model Performance

The selection of an activation function is also crucial during the training process, since it has a significant impact on how well the same model performs. During the experiment, we replaced the *Leaky ReLU* activation function with three activation functions, *ReLU*, Sigmoid, and Tanh. The accuracy and loss values employing the two activation functions Sigmoid and Tanh differed significantly from those obtained with *ReLU*, as shown in Table 6. In contrast to the *ReLU* activation function, the *Leaky ReLU* function showed an improvement in accuracy of 0.63 and a reduction in loss value of 0.023. There are two reasons for this result: (1) the *Leaky ReLU* activation function could transmit negative weights, which makes the model more capable of pushing information; (2) the *Leaky ReLU* activation function transmitted more detail information, such as texture, line, and color of ginseng, to extract feature detail information that was not easily extracted by the network, which led to a substantial improvement in model performance.

**Table 6.** Experimental comparison results of different activation functions.

| Activate Function | Accuracy/% | Loss |
|:---:|:---:|:---:|
| Sigmoid | 63.84 | 0.308 |
| Tanh | 75.76 | 0.216 |
| *ReLU* | 96.76 | 0.058 |
| *Leaky ReLU* | 97.39 | 0.035 |

### 4.4. Effect of the Loss Function on the Model Performance

To investigate the effect of various loss functions on the model performance, we selected a cross-entropy loss function for comparison. Although the cross-entropy loss function convergence effect is very good, but it still has limitations: in real applications, containing the complex background information of an image—such as, in this case, the fact that the number of ginseng pixels was much smaller than the number of background pixels—can lead to prevailing background information in the cross-entropy loss function composition; therefore, the network model is clearly biased, resulting in a decline in recognition effect. As can be seen in Table 7, the accuracy of the model utilizing the focal loss function increased to some extent both before and after the improvement, with a significant decrease in the loss values. This indicated that the introduction of a focal loss function could adequately address the problem of sample imbalance in the dataset and has a positive influence on the final classification of the model.

**Table 7.** Experimental comparison of the results obtained with different loss functions.

| Model | Loss Function | Accuracy/% | Loss |
|:---:|:---:|:---:|:---:|
| ResNet50 | Cross entropy Loss | 94.78 | 0.176 |
| | Focal Loss | 94.94 | 0.066 |
| Our model | Cross entropy Loss | 97.22 | 0.119 |
| | Focal Loss | 97.39 | 0.035 |

### 4.5. Model Evaluation

In image classification, the prediction results of a classification model are frequently represented as a confusion matrix, from which the metrics Accuracy, Recall, and Precision are derived:

$$Accuracy = \frac{TP}{TP + FN + FP + TN} \tag{8}$$

$$Recall = \frac{TP}{TP + FN} \tag{9}$$

$$Precision = \frac{TP}{TP + FP} \tag{10}$$

where *TP* (true positive) is the number of ginseng specimens accurately identified, *FP* (false positive) is the number of ginseng specimens incorrectly identified, *FN* (false negative) is the number of ginseng specimens identified as other types of ginseng, and *TN* (true negative) is the number of other ginseng specimens accurately identified.

In the confusion matrix in Figure 9, 1 represents the principal ginseng, 2 the first-class ginseng, 3 the second-class ginseng, and 4 the inferior ginseng; the horizontal coordinates represent the true category, and the vertical coordinates represent the predicted category. According to the confusion matrix in Figure 9, the ResNet50 model has an overall accuracy of 94.94% for the test set, an overall recall of 94.37%, and an accuracy of 94.99%. The overall accuracy of our proposed model's test set was 97.39%, while its overall recall was 97.03%, and its accuracy was 97.29%.

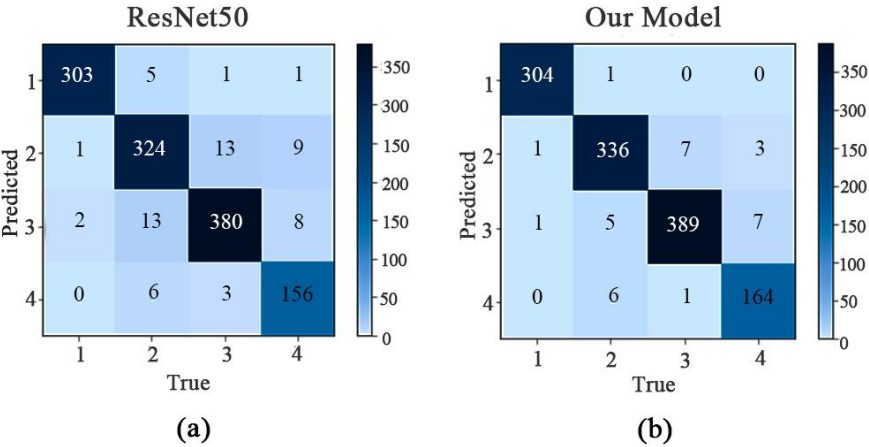

(a)  (b)

**Figure 9.** Confusion matrix before and after the model improvement. (**a**) ResNet50 confusion matrix; (**b**) our model confusion matrix. Note: 1 is principal; 2 is first-class; 3 is second-class; 4 is inferior. The x-axis is the real category, and the y-axis is the predicted category.

In Table 8, the improved model enhanced the recall of all four types of ginseng to different degrees compared to the original model. However, the model still showed a large error in the identification of "inferior" ginseng; thus, improving the identification rate of inferior ginseng will be the focus of our future research.

**Table 8.** Recall and accuracy of ginseng classification by class before and after model improvement.

| Level | Model | Recall/% | Precision% |
|---|---|---|---|
| Principal | ResNet50 | 99.02 | 97.74 |
| | Our model | 99.35 | 99.67 |
| First-class | ResNet50 | 93.10 | 93.37 |
| | Our model | 96.56 | 96.83 |
| Second-class | ResNet50 | 95.72 | 94.29 |
| | Our model | 97.98 | 96.77 |
| Inferior | ResNet50 | 89.66 | 94.55 |
| | Our model | 94.25 | 95.91 |

## 5. Discussion

Ginseng is a traditional Chinese medicine also used as food and is also a common health care medicine, very appreciated by people. The identification of the quality of ginseng is therefore of paramount importance and is an important area of concern for the majority of researchers in this field, today. The traditional methods of identifying the appearance and quality of ginseng do not offer advantages in the age of artificial intelligence; therefore, after learning about the achievements of convolutional neural networks in the field of image processing, we decided to use ResNet50 as a base model to improve the

evaluation of ginseng quality based on the small differences in features and details found in ginseng plants of different levels of quality. However, as can be seen from the confusion matrix with the original model, inferior ginseng had the lowest recognition accuracy, which also reflects the need to further improve the model's ability to extract features in order to provide better recognition results when using a ginseng dataset with highly similar specimens. We subjected the improved model to multiple ablation experiments on the ginseng dataset, and the results showed that the improved ResNet50 outperformed other models mainly in terms of accuracy and loss values and offered greater advantages in terms of convergence speed and stability; it also showed a slightly faster training time per round than the other models. The model performance was further improved with a recognition accuracy of 97.39%. We also proved through experiments that the method could improve the identification accuracy without damaging the appearance of the ginseng specimens and basically met the identification needs.

## 6. Conclusions

The focus of our future research is on the further optimization of this model structure, with reference to the needs of realistic applications, focusing on reducing the weight of the model so that it could still achieve effective classification tasks with fewer parameters. As this study was only about the classification of white ginseng specimens of different quality, we will next construct a dataset for ginseng from different origins and different years to analyze the effect of the appearance characteristics of ginseng from different backgrounds on its classification. In addition, we will deploy the model on mobile devices for use in public promotion, market supervision, and industrial production. We will make it easier and faster for people to identify the quality of ginseng, thus improving public awareness, market supervision, and industrial production efficiency. This will provide a better solution to the problem of identifying the appearance of ginseng quality in theory and practice.

In this paper, the traditional ResNet50 network was improved according to the characteristics of ginseng. Our conclusions are as follows.

First of all, we constructed a ginseng dataset and classified the data to identify samples of four different quality levels: principal, first-class, second-class, and inferior ginseng. Data augmentation was then performed on the dataset to extend it. Then, we replaced the traditional activation function *ReLU* with the *Leaky ReLU* activation function to enhance the expressiveness of the model. We also introduced the ECA mechanism module on the residual block to increase the model's sensitivity to ginseng pixels and better capture the ginseng specimens' features. Additionally, the focal loss function was introduced to balance the dataset, and the idea of transfer learning was used to train the model. Finally, our proposed model showed greater advantages in different aspects compared to the classical convolutional neural network models Vgg16, GoogLeNet, ResNet50, and Densenet121. Many comparison experiments clearly showed that the method proposed in this paper had a beneficial impact on the model performance. Compared with the original model, the accuracy and loss value of the improved model were the best. In addition, with the significant improvement of the model performance indicators, the accuracy and overall recall rate were also improved. These results also validated the effectiveness and feasibility of the improvements to the original model. In future studies, we plan to further optimize the model and combine it with a mobile terminal to provide technical support for ginseng appearance quality recognition.

**Author Contributions:** Conceptualization, W.L. and D.L.; methodology, D.L. and X.P.; software, D.L. and X.P.; validation, D.L., X.P. and Y.L.; formal analysis, W.L. and L.Z.; investigation, Y.L.; resources, D.L. and X.P. and W.L.; data curation, X.P. and Y.L.; writing—original draft preparation, L.M., W.L., D.L., X.P., Y.L. and L.Z.; writing—review and editing, L.M., W.L., D.L., X.P., Y.L. and L.Z.; visualization, L.M., W.L., D.L., X.P., Y.L. and L.Z.; supervision, D.L.; project administration, D.L.; funding acquisition, L.M., W.L., D.L. and L.Z. All authors have read and agreed to the published version of the manuscript.

**Funding:** This research was funded by the National Natural Science Foundation of China (No. 61801439); the Department of Science and Technology of Jilin Province (20210204050YY); Jilin Provincial Education Department Scientific Research Project (JJKH20210747KJ); Jilin Provincial Environmental Protection Department Project (202107); Jilin Provincial Middle and Young Leaders Team and Innovative Talents Support Program (No. 20200301037RQ).

**Institutional Review Board Statement:** Not applicable.

**Informed Consent Statement:** Not applicable.

**Data Availability Statement:** All relevant data are included in the manuscript. Raw images are available on request from the corresponding author.

**Conflicts of Interest:** The authors declare no conflict of interest.

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
