# Peer review of "A Grading Method of Ginseng (Panax ginseng C. A. Meyer) Appearance Quality Based on an Improved ResNet50 Model"

_agronomy, doi:10.3390/agronomy12122925_

Round 1
Reviewer 1 Report
This article introduces a grading quality method for ginseng. Although the article provides an interesting case, its structure and format should be improved before it can become suitable for publication. One major area that needs to be improved is the language and grammar. Also, there are some issues with the images taken for the experiment. For example in Figure 1, the lighting (and the resulting shadows) are not consistent. The authors should explain why the images are taken with different light angles. In terms of the model, although the loss for the model is lower than the previous models, the amount of improvement is not very high, in other words, the authors should be able to find other angles than improving the accuracy such lower computational requirements etc why people should invest in applying your model? I don much information on comparing the computational requirements of these models. The discussion section is very short and does not reflect the connection to the results that other researchers have obtained. Authors are also encouraged to publish their code and data as separate articles so other researchers could also benefit from it.
Some specific comments:
Line 31: Scientific names should be in Italic
Lines 32-33: Please provide a reference for the benefits of ginseng here
Line 37: If Empirical identification is a scientific method, provide a reference for it so the audience can read and understand more.
Line 42: ‘suffer’ is not a suitable word. You can damaged instead.
Line 48: primarily written and not very operational: you mean they are not operationally feasible?
Line 53: “Deep learning uses a method of thinking similar to the human brain” this is a wrong claim, DL doesn’t really think and Neural Network methodology does not really reflect how prediction is made by human brain.
Table 1: Please organise the information in a better way, the pieces of text often overlap
Lines 110-113: this sentence is very long. Please try and shorten it.
Line 127: “To eliminate the possibility of experimental results”: what this means?
Line 156: Please add complete definition the first time ReLU is mentioned in the text.
Reviewer 2 Report
The paper deals with a topic of interest such as the remote evaluation of essences with high added value such as ginseng, with a targeted test protocol, setting and identification, discussion of the results; the approach followed by the authors does not sufficiently characterize the connection between the evaluated essence and the system used for the acquisition, correction and training of the artificial intelligence system, the methodology thus described could be used for many other crop or weed, the work should be deep improved and integrated trying to link methodology, software and training for the specific purpose, perhaps also highlighting the specificities of the examined crop as seen for example in weed recognition models for interrow control. For example in an innovative research paper on valuable crops the discussions paragraph cannot be too obvious, generic, limit to 12-13 lines, it is self-limiting of the work done by the Authors.
Reviewer 3 Report
The paper evaluates a deep-learning-based quality grading method for ginseng based on appearance quality identification. The methodology has used Leaky ReLU and ECA module to the residual block, which allowed the researcher to focus on the input object to capture more precise and detailed features. Methods used are robust, in my opinion, and reported results comparatively outperformed other techniques available. I don’t notice any shortcomings in the research except for editorial revisions. I recommend you expand your discussion by reflecting on existing knowledge and the importance of your research findings.
Specific comments.
36 – Reference needs adjustment to reflect authors names to properly read the sentence.
201-202 – sentence does not read well. Revise for better flow.
236 – Figure 6 caption needs to be self-explanatory of the content. Expand all abbreviations with full names.
237-251 – you have used conv layer and conv block ; abbreviated forms throughout the paragraph. I suggest you introduce the abbreviation where its used first and then consistently use abbreviated form throughout the text. Pay attention to capitalization as well.
303 – Include the expanded form of all abbreviations in the table caption.
354 – include 1 represents for the principal, 2 for the first-class, 3 for the second-class and 4 for the inferior in the figure caption
361 – your research and methods are promising. I would encourage you to expand the discussion to reflect your findings in relation to what is already known.
Round 2
Reviewer 1 Report
The paper is much improved.
Author Response
感谢您对文章的宝贵意见,这使我们的文章更加规范。同时,感谢您对我们修订稿件的肯定。
亲切问候!
Reviewer 2 Report
The work has been well improved and is now more linear and smooth, the discussions could be summarized while the conclusions should be streamlined, the values are usually not reported, some of them should be brought to the appropriate paragraph, the commitment of the Authors is evident
Author Response
请参阅附件。
